# Screening of Endophytic Fungi in Locoweed Induced by Heavy-Ion Irradiation and Study on Swainsonine Biosynthesis Pathway

**DOI:** 10.3390/jof8090951

**Published:** 2022-09-10

**Authors:** Yanan Mo, Zhen Yang, Baocheng Hao, Feng Cheng, Xiangdong Song, Xiaofei Shang, Haoxia Zhao, Ruofeng Shang, Xuehong Wang, Jianping Liang, Shengyi Wang, Yu Liu

**Affiliations:** 1Key Laboratory of New Animal Drug Project, Lanzhou 730050, China; 2Key Laboratory of Veterinary Pharmaceutical Development, Ministry of Agriculture and Rural Affairs, Lanzhou 730050, China; 3Lanzhou Institute of Husbandry and Pharmaceutical Sciences, Chinese Academy of Agriculture Sciences, Lanzhou 730050, China; 4Key Laboratory of Plant Resources and Chemistry in Arid Regions, Xinjiang Technical Institute of Physics and Chemistry, Chinese Academy of Sciences, Urumqi 830000, China; 5College of Veterinary Medicine, Gansu Agricultural University, Lanzhou 730050, China

**Keywords:** *Alternaria* section *Undifilum oxytropis*, swainsonine, heavy-ion irradiation, metabonomics, biosynthesis pathway

## Abstract

Swainsonine (SW) is a substance with both animal neurotoxicity and natural anticancer activity produced by the metabolism of endophytic fungus *Alternaria* section *Undifilum oxytropis* of locoweed. This paper produced SW by fermentation of the endophytic fungus A. *oxytropis* of locoweed and obtained the optimal ultrasonic-assisted extraction process of SW by the response surface methodology. Meanwhile, four mutant strains with significant and stable SW-producing properties were screened out after the mutagenesis of *A. oxytropis* by heavy-ion irradiation. Of these, three were high-yielding stains and one was a low-yielding strain. In addition, through the analysis of metabolomics studies, it was speculated that the different SW production performance of the mutant might be related to the biosynthesis and utilization of L-lysine, L-2-aminoadipate-6-semialdehyde, etc. These results laid the foundation for the expansion of SW production, artificial construction of low-toxic locoweed and clarification of the SW biosynthesis pathway in *A. oxytropis*.

## 1. Introduction

Locoweed is an international collective term for poisonous plants in the genera *Astragalus*, *Echinotropis* and *Sophora*. Long-term feeding on locoweed will cause nerve dysfunction, reduced reproductive capacity, growth inhibition and even death of livestock. The main toxin of locoweed is swainsonine (SW), produced by endophytic fungi metabolism. SW is a kind of indolizidine alkaloid, which was first isolated from the poisonous legume plant *Swainsona canescen* by Australian scholar Colegate [1]. The researchers found that SW competitively inhibits mannosidase activity, triggering enzyme dysfunction and the accumulation of oligosaccharides, leading to vacuolar degeneration in different cells, especially nerve cells [2]. Meanwhile, SW can promote the apoptosis of tumor cells by affecting the synthesis of various carbohydrate, glycoproteins and glycolipids. Therefore, SW is also a natural anti-tumor drug with broad clinical medicinal prospects [3]. SW is currently obtained from three sources: plant extraction, artificial synthesis and fungal fermentation. Up to now, the price of pure SW has been very high, mainly because its chemical synthesis process is cumbersome and difficult to purify, and its extraction from plants is complicated and inefficient, which restricts the research and development of SW as an anti-tumor drug [4]. It has been reported that abundant SW was found in the *Alternaria* section *Undifilum oxytropis* (*A. oxytropis*), which was the endophytic fungi of locoweed [5]. In addition, fungal fermentation has the advantages of being low cost, easy to control and environmentally friendly. Therefore, the method of extracting SW by fermentation of *A. oxytropis* has great potential in the field of biopharmaceutics.

Stable microbial mutants with desired phenotypes could be obtained by mutation breeding by altering the genome, which has been widely used in the bioindustry [6]. Heavy-ion beam is a new source of radiation used in mutagenesis technology. Compared with other traditional mutagenesis methods, such as ultraviolet, C-rays and chemical mutagenesis, heavy-ion beam irradiation mutagenesis has many advantages, including high relative biological effect, high energy transmission line density, high mutagenesis rate, difficult recoverable mutation and broader breeding mutation spectrum in microbial and plant environments [7]. Currently, heavy-ion beam is used as an effective method for the mutagenesis breeding of new microbial and plant varieties [8]. Many favorable plant varieties with desired phenotypes have been obtained through heavy-ion mutagenesis breeding experiments on plants, such as wheat, tomato, maize, forage, etc. [9]. Inspired by the advantages of plant breeding and heavy ion mutagenesis technology, the use of heavy ion irradiation for microbial mutagenesis breeding has attracted the extensive attention of researchers. So far, with the help of heavy-ion irradiation, more than 20 promising microbial strains have been cultivated, including bacteria, fungi, and microalgae, etc. Some mutated microbial strains via heavy-ion irradiation, such as *Streptomyces avermitilis*, *Saccharomyces cerevisiae* and *Aspergillus niger*, have been transferred to industrial-scale production [8].

Extensive studies have been conducted worldwide on the initial steps of the SW biosynthetic pathway in the endophytic fungi of locoweed. SW is a metabolite of lysine through saccharopine, 1-piperidine-6-carboxylate and pipecolate [10]. Li et al. constructed the gene knockout mutant M1 of *A. oxytropis* and sequenced the transcriptome of wild-strain OW7.8 and M1 at the first time [11]. There were 41 unigenes possibly related to the biosynthesis of SW identified by data analyzing. The biosynthesis pathway of SW in *A. oxytropis* was speculated to include two branches of Δ1-piperideine-6-L-carboxylate (P6C) and Δ1-piperideine-2-carboxylate (P2C). Moreover, by constructing saccharopine reductase gene (Sac) complementary strain C1, researchers found that C1 had a significantly higher SW level, compared with Sac knockout strain M1 and wild strain OW7.8, which might be due to the strong promoter in C1 that overexpressed Sac, thus promoting the generation of SW in C1. These results proved that Sac plays an important role in SW biosynthesis [12]. However, the detailed biosynthetic pathway of SW in *A. oxytropis* is poorly understood currently.

In this study, the ultrasonic-assisted extraction process of SW from the mycelia of *A. oxytropis* was optimized by the response surface methodology. Then, mutants with different levels of SW production were screened by the optimized extraction process. The mutants with high SW yield were cultured by fermentation to provide a sufficient SW source for its anti-tumor research, while the mutants with low SW yield were screened to construct the low-toxic/non-toxic locoweed, which could fundamentally solve animal locoweed poisoning disease and comprehensively utilize and manage the locoweed resources. In addition, the metabolomics method was used to study three kinds of strains with different SW production performance results to explore the SW biosynthetic pathway in *A. oxytropis*, which provided a certain theoretical basis for the transformation and utilization of endophytic fungus in locoweed.

## 2. Materials and Methods

### 2.1. Material Preparation

*Alternaria* section *Undifilum oxytropis* (number: CICC2493) was taken from the China Industrial Microbial Strain Preservation Center. Swainsonine standard was purchased from Yangling Tianli Biotechnology Co., Ltd. (Xianyang, China).

### 2.2. Reagents Preparation

Formic acid (LC-MS), methyl alcohol (LC-MS) and acetonitrile (LC-MS) were purchased from Thermo. Water (LC-MS) was purchased from Merck. Methyl alcohol (analytically pure) and formic acid (analytically pure) were bought from local suppliers. The 1 × Phosphate buffer solution (PBS) was purchased from Solarbio Science & Technology Co., Ltd. (Beijing, China). Potato dextrose agar (PDA) medium and potato dextrose broth were obtained from Huankai Microbial.

### 2.3. Optimization of Extraction Technology of SW from A. Oxytropis by Response Surface Methodology

#### 2.3.1. Preparation of Dry Powder of *A. oxytropis* and the Extraction of SW

The preparation of the dry powder of *A. oxytropis* and the extraction of SW were performed following the method described by Song et al. with modifications [13]. *A. oxytropis* was inoculated in PDA medium and cultured at 25 °C for 30 days until the strain recovered. Transferred 500 μL fungi suspension was made by grinding mycelium with sterile water into a conical flask containing 150 mL of sterilized potato dextrose broth, and the bottle was placed in a constant temperature shaker at 27 °C and 130 rpm for continuous cultivation for 24 days. After the cultivation, the filtrated, rinsed, dried mycelium pellets were ground into dry powder with liquid nitrogen. Then, 0.2 g dry mycelium powder was weighed and added into 6 mL of methanol. Ultrasonic extraction was performed at 60 °C for 40 min, centrifugation was performed at 3500 rpm for 5 min, and the supernatant was collected. This step was repeated three times. All of the supernatant was collected, dried in a water bath at 70 °C, redissolved with 5 mL methanol, and filtered by a 0.22 μm organic phase needle filter. The obtained solvent was then diluted with methanol (LC-MS) and transferred to an injection bottle for later use.

#### 2.3.2. Determination of SW Content

The content of SW in *A. oxytropis* was determined by UPLC-MS/MS. A total of 1 mg of SW standard substance was accurately weighed and prepared into the 100 μg/mL reserve solution with methanol (LC-MS). A series of solutions with mass concentrations of 350, 300, 200, 150, 100 and 50 ng/mL were prepared by successively diluting methanol (LC-MS) to make the standard curve. The chromatographic conditions were as follows: KINETEX C_18_ (100 × 2.1 mm, 2.6 μm); mobile phase, 0.1% formic acid aqueous solution (A), methanol (B); mobile phase ratio was 30% of A, 70% of B; flow rate, 0.22 mL/min; column temperature, 40 °C; and injection volume, 3 μL. MS conditions were as follows: ion source, ESI+; scanning mode, positive ion mode, multiple reaction monitoring (MRM); curtain gas (CUR), 20 psi; collision gas (CAD), medium; IonSpray voltage (IS), 5500 V; temperature (TEM), 550 °C; and GS1 and GS2, 344.7 kPa. The qualitative and quantitative ion pairs were *m/z* 174.1/156.1; the optimized declustering potential (DP) was 84; the entrance potential (EP) was 10; the collision energy (CE) was 19; the collision cell exit potential (CXP) was 13; and the retention time (RT) was 50 msec. 

#### 2.3.3. Methodology Validation

The methodology validation was performed based on an earlier reported method with some modifications [14], including a precision test, repeatability test, stability test and sample addition recovery test.

Precision test: The SW standard solution was continuously injected by UPLC-MS/MS six times, the SW ion pair peak area was measured, and its relative standard deviation (RSD) was calculated. If RSD < 3%, it indicated that the instrument precision was good and met the experimental requirements. The RSD calculation formula is as follows: RSD = Standard deviation (SD)/Average value (X)

Repeatability test: six samples of the same batch of dry mycelium powder were respectively measured to prepare the test solution, 0.2 g per sample. UPLC-MS/MS was used to measure the peak area of SW ion pair and calculate the content of SW in the sample and its RSD. If RSD < 3%, it indicated that the method had good repeatability and met the experimental requirements.

Stability test: UPLC-MS /MS was used to measure the peak area of SW ion pair of the same sample solution after 0, 1, 2, 4, 8, 12, and 24 h preparation, and the RSD was calculated. If RSD < 3%, it indicated the method had good stability and met the experimental requirements. 

The sample addition recovery test: Six parts of mycelium dry powder with known SW content (87.94 μg/g) were placed into 15 mL centrifuge tubes respectively, 0.2 g each part. The test solution was prepared by adding 1 mL of 20 μg/mL reference solution. The average recovery rate and RSD of SW were determined by UPLC-MS/MS.

#### 2.3.4. Single Factor Experiment

The effects of extraction time, extraction temperature, concentration of formic acid in solvent and the material to liquor ratio on the yield of SW were investigated. Six levels of each factor were set for the single factor experiment. The experimental design was as follows: (1)Comparison of the extraction times: the concentration of formic acid in solvent was 0, the material-to-liquid ratio was 1: 30 g/mL, the extraction temperature was 60 °C, and the extraction times were set to 20, 40, 60, 80, 100 and 120 min, respectively.(2)Comparison of the extraction temperatures: the concentration of formic acid in solvent was 0, the extraction time was 40 min, the material-to-liquid ratio was 1: 30 g/mL, and the extraction temperatures were set to 30, 40, 50, 60, 70 and 80 °C, respectively.(3)Comparison of the concentrations of formic acid in solvent: the material-to-liquid ratio was 1: 30 g/mL, the extraction temperature was 60 °C, the extraction time was 40 min, and the concentrations of formic acid in solvent were set to 0, 0.5%, 1%, 2%, 4% and 6%, respectively.(4)Comparison of the material-to-liquid ratio: the concentration of formic acid in the solvent was 0, the extraction temperature was 60 °C, the extraction time was 40 min, the material-to-liquid ratios were set to 1:10, 1:15, 1:20, 1:25, 1:30 and 1:35 g/mL, respectively. The experiment was repeated three times in each group to determine the content of SW and take the average value.

#### 2.3.5. Optimization of SW Extraction Process by Response Surface Methodology

According to the results of single factor experiment, four factors, including extraction time (A), extraction temperature (B), formic acid concentration (C) and material-to-liquid ratio (D), were selected as variables. Three-level response surface experiment of four factors designed by Design-Expert 8.0.6 was carried out, −1, 0 and 1 represented variable levels. The real value and coding level of independent variables are shown in Table 1.

#### 2.3.6. Verification Test

Five samples were accurately weighed, each 0.2 g, and SW was extracted according to the modified extraction conditions and the extraction rate of SW was calculated. 

### 2.4. Mutagenesis of A. Oxytropis by Heavy-Ion Irradiation Technology

#### 2.4.1. Treatment of *A. oxytropis* Suspension by Heavy-Ion Irradiation

*A. oxytropis* was cultured on PDA at 25 °C for 30 days. The colonies with good growth were selected and placed in the mycelium grinder. We added 10 mL of PBS (sterilization) to grind. The mycelia were filtered and removed with three layers of lens wiping paper, and the filtered fungi solution was transferred into a 10 mL EP tube for later use. A total of 2 mL suspension was added to 35 mm irradiation dish and irradiated by 80 MeV/u carbon ions with the LET of 40 Kev/μm, which was provided by the Heavy-Ion Research Facility in Lanzhou (HIRFL), Institute of Modern Physics, Chinese Academy of Sciences. The irradiation doses were set as 40, 60, 80, 100, 120 and 140 Gy, and three replicates were set for each dose. The lethality rate after irradiation was calculated according to the following formula: Lethality rate (%) = (1 − number of growing colonies in irradiation group/number of growing colonies in blank control group) × 100%.

#### 2.4.2. Determination of SW Content and Screening of Mutagenic Strains

The mutant strains treated by heavy-ion irradiation were isolated and numbered. The strains with good growth were selected for the fermentation culture. The optimized method was used to extract SW from mycelia, and UPLC-MS/MS technology was used to determine the content of SW in mutant strains. The high and low yielding SW strains were screened by the different content of SW in the mycelia of mutant strains. 

#### 2.4.3. Observation of Mutant Colony Morphology and Stability Test of SW Production

The mutated strains with significant changes in SW yield were subcultured for five consecutive generations, and each generation was cultured for 24 d. The SW content in mycelia was detected to evaluate the stability of SW production at the end of the culture. Meanwhile, the growth status and colony morphology of mutants on PDA and potato dextrose broth were observed and recorded. 

### 2.5. Metabolomics Studies

#### 2.5.1. Metabolites Extraction

Mutants 61, 70 and the original strain were fermented and cultured for 24 days. After that, the mycelium pellets were collected by filtration, and cleaned with sterile distilled water three times. Then the samples were placed into EP tubes and put into liquid nitrogen for quick freezing for 5 min after resuspended with prechilled 80% methanol. The frozen samples were melted on ice, vortexed for 30 s, sonificated for 6 min, and centrifuged at 5000 rpm for 1 min at 4 °C. The supernatant was freeze dried and dissolved with 10% methanol. Finally, the solution was injected into the LC-MS/MS system for analysis [15,16].

#### 2.5.2. HPLC-MS/MS Analysis

LC-MS/MS analyses were performed using an ExionLC™ AD system (SCIEX) coupled with a QTRAP^®^ 6500+ mass spectrometer (SCIEX) in Novogene Co., Ltd. (Beijing, China). Samples were injected onto a Xselect HSS T3 (2.1 × 150 mm, 2.5 μm) using a 20-min linear gradient at a flow rate of 0.4 mL/min under the positive/negative polarity mode. The eluents were 0.1% formic acid–water (eluent A) and 0.1% formic acid–acetonitrile (eluent B) [17]. The solvent gradient was set as follows: 2% B, 2 min; 2–100% B, 15.0 min; 100% B, 17.0 min; 100–2% B, 17.1 min; and 2% B, 20 min. QTRAP^®^ 6500+ mass spectrometer was operated in positive polarity mode with curtain gas of 35 psi, collision gas of medium, IonSpray Voltage of 5500 V, temperature of 550 °C, ion source gas of 1:60, and ion source gas of 2:60. QTRAP^®^ 6500+ mass spectrometer was operated in negative polarity mode with curtain gas of 35 psi, collision gas of medium, IonSpray voltage of −4500 V, temperature of 550 °C, ion source gas of 1:60, and ion source gas of 2:60.

#### 2.5.3. Metabolites Identification and Quantification 

Based on Novogene in-house database, the multiple reaction monitoring mode (MRM) was used to detect the experimental samples. The metabolites were quantitatively analyzed according to Q3 (daughter ion), and qualitatively analyzed according to Q1 (parenion), Q3 (daughter ion), RT (retention time), DP (declustering potential) and CE (collision energy). The data files generated by HPLC-MS/MS were processed by SCIEX OS Version 1.4, including the integration and correction of the chromatographic peaks. The main parameters were set as follows: minimum peak height, 500; signal/noise ratio, 5; and Gaussian smooth width, 1. The area of each peak represents the relative content of the corresponding substance.

#### 2.5.4. Data Analysis

Principal components analysis (PCA) and partial least squares discrimination analysis (PLS-DA) were performed at metaX, which was a flexible and comprehensive software for metabolomics data processing) [18,19], to obtain the values of VIP (variable importance in the projection). The statistical significance (*p*-value) and fold change (FC) of each metabolite between the two groups were calculated based on univariate analysis (*t*-test). The metabolites with VIP > 1, *p*-value < 0.05 and FC ≥ 1.2 or FC ≤ 0.833 were considered to be differential metabolites [20,21,22]. Volcano plots, plotted with ggplot2 in R language, combined with VIP, FC (log_2_) and *p*-value (−log_10_) parameters of metabolites, could be used to screen metabolites of interest. Clustering heat maps were plotted by Pheatmap package in R language, and the data of the intensity areas of differential metabolites were normalized using z-scores. The functions and metabolic pathways of these metabolites were studied using the KEGG database (http://www.genome.jp/kegg/, accessed on 12 August 2022). When x/n > y/n, the metabolic pathway was considered to be enriched, when *p*-value < 0.05, the metabolic pathway was considered to be statistically significant enriched.

### 2.6. Statistical Analysis

Experimental data was expressed as the mean ± SE of three independent experiments and analyzed by ANOVA analysis of variance using IBM SPSS Statistics 23. *p*-values less than 0.05 were considered statistically significant.

## 3. Results

### 3.1. Optimization of Extraction Technology of SW from A. Oxytropis by Response Surface Methodology

#### 3.1.1. Standard Curve of SW, Lower Limit of Detection, Lower Limit of Quantification

The standard curve of SW obtained by UPLC-MS/MS was shown in Figure 1. The linear regression equation is as follows: *Y* = 2.58907 × 10^5^·*X* + 3.38105 × 10^6^ (R^2^ = 0.9991)

The linear concentration range was 50–350 ng/mL. The lower limit of quantification (S/N 10:1) and the lower limit of detection (S/N 3:1) were 6.22 ng/mL and 1.87 ng/mL, respectively, which were both low enough to meet the experimental requirements.

The calculation formula of the SW extraction quantity is as follows:SW extraction quantity = *N*·*C*·*V*/*W*

where *N* represents the dilution ratio, *C* is the concentration of SW measured, *V* is the volume of the sample solution during the extraction of redissolution, and *W* is the weight of dry mycelia powder sample during the extraction of *A. oxytropis*.

**Figure 1 jof-08-00951-f001:**
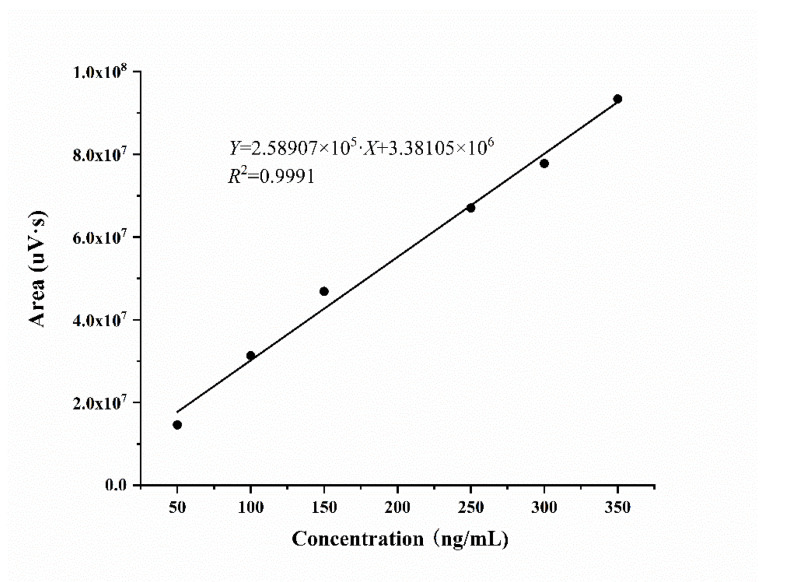
The standard curve of SW.

#### 3.1.2. Methodology Validation

The SW standard solution (200 ng/mL) was continuously measured six times by UPLC-MS /MS, and the peak areas of SW ion pair were 6.703 × 10^7^, 6.669 × 10^7^, 6.801 × 10^7^, 6.598 × 10^7^, 6.724 × 10^7^, 6.738 × 10^7^, respectively. The RSD was 1.02%. Results of repeatable experimental are shown in Table 2. The average content of SW in the sample was 88.024 μg/g, and the RSD was 2.12%. The peak areas of SW ion pair measured at 0, 1, 2, 4, 8, 12 and 24 h after preparation of the same sample solution were 4.837×10^7^, 4.729×10^7^, 4.906 × 10^7^, 4.859 × 10^7^, 4.876 × 10^7^, 4.896 × 10^7^ and 4.796 × 10^7^ respectively, and the RSD was 1.29%. The average recovery rate of SW in the spiked recovery test was 94.71%, within the limited range, and the RSD was 2.82% (Table 3). In summary, RSD values of all experimental results were less than 3%, indicating that this method met the experimental requirements, and the next test could be carried out after verification.

#### 3.1.3. Single-Factor Experiment

The results of the single-factor experiment are shown in Figure 2. On the premise that there was only one variable, the amount of SW extraction increased first and then decreased with the increase in the value of this variable. The extraction amount of SW reached the peak when extraction time was 80 min, temperature was 40 °C, formic acid concentration was 1% and material-to-liquid ratio was 1:25 g/mL. Therefore, the extraction time range of 60~120 min, the temperature of 30~50 °C, the formic acid concentration of 0~2%, and the material-to-liquid ratio of 1:20~1:30 g/mL were suitable for subsequent research. 

#### 3.1.4. Optimization of SW Extraction Process by Response Surface Methodology

The experimental scheme obtained by Design-Expert 8.0.6 and corresponding results are shown in Appendix A. Through subsequent analysis, the quadratic multiple regression equation of the influence of interaction of various factors on the amount of SW extraction was obtained as follows: Y = 210.56 + 1.11A + 4.17B − 7.2C − 0.54D − 2.78AB + 5.81AC − 1.39AD + 3.97BC + 4.99BD + 2.29CD − 15.49A^2^ − 12.63B^2^ − 18.12C^2^ − 16.65D^2^
where Y is the extraction amount of SW, A is the extraction time, B is the extraction temperature, C is the concentration of formic acid, and D is the material-to-liquid ratio. The variance analysis data of the constructed response surface model are shown in Table 4. The model had statistical significance (*p* < 0.05) and low error (*p* > 0.05 for Lack of Fit), indicating that the model was successfully constructed and could be used to analyze the ultrasound-assisted extraction process of SW. According to *F*-value, formic acid concentration had the greatest influence on the extraction rate of SW (*p* < 0.05), followed by ultrasonic temperature and time, and the material-to-liquid ratio had the least influence.

Figure 3 shows the response surface model and contour diagram of the effect of the different factors interaction on the extraction rate of SW. It can be seen that the effects of extraction time, temperature and material–liquid ratio on the SW extraction rate are not significantly different when the three factors interact in pairs. The effect of the extraction temperature was slightly larger than that of the extraction time, and the effect of time was slightly larger than that of the material–liquid ratio. The change trend of SW extraction rate was to increase first and then decrease with the increase in their variables. The contour lines were denser, and the SW extraction rate changed greatly when the extraction time was 60~90 min and the ultrasonic temperature was 30~40 °C. When the formic acid concentration was a variable, the interaction between it and the other three factors showed significant differences on the effect of the SW extraction rate. When the concentration of formic acid was 1~2%, the contour lines were denser, indicating this range had the greatest influence on the extraction rate of SW.

#### 3.1.5. Verification Test

The optimal extraction conditions were a time of 89.66 min, ultrasonic temperature of 41.36 °C, formic acid concentration of 0.81%, and material–liquid ratio of 1: 24.96 g/mL through Box–Behnken analysis. The predicted extraction amount of SW was 211.504 μg/g. According to the actual operation situation, the extraction conditions changed to the extraction time of 90 min, temperature of 41 °C, formic acid concentration of 0.8%, and material–liquid ratio of 1:25 g/mL. The amounts of SW extracted from *A. oxytropis* are shown in Table 5 after actual operation. The average content was 220.572 μg/g, which was not much different from the predicted value of the model. The RSD was 2.02%, less than 3%, indicating that the optimal extraction conditions of the response surface method were accurate and reliable. 

### 3.2. Mutagenesis of A. Oxytropis by Heavy-Ion Irradiation Technology

#### 3.2.1. Lethality Rate of *A. oxytropis* Strain Induced by ^12^C^6+^ Heavy ion Beam Irradiation

The lethality rate of *A. oxytropis* strain irradiated with different doses of ^12^C^6+^ heavy ion beam was shown in Table 6. The fatality rate generally showed an upward trend. When the irradiation dose was 40 Gy, the average fatality rate was 69.23%; when the irradiation dose increased to 60 Gy, the average fatality rate decreased slightly to 66.67%, and then the fatality rate continued to rise.

#### 3.2.2. Determination of SW Content and Screening of Mutagenic Strains

The mutant strains mutated by heavy-ion beam irradiation were isolated and numbered. The strains with good growth were selected for fermentation culture, and their numbers were 2, 55, 59, 61, 63, 64, 68 and 70, respectively. UPLC-MS/MS was used to quantitatively analyze SW in mycelia of different mutants, and the high-producing and low-producing SW strains were screened by the different SW content in mycelia of mutants. Figure 4 showed the content of SW produced by mutants after mutagenesis, and the specific results were shown in Appendix A. Group C was the original strain of *A. oxytropis*. The content of SW in mycelia of mutated strains numbered 55, 63, 64 and 70 were 317.79 μg/g, 263.57 μg/g, 387.44 μg/g and 469.64 μg/g, respectively. Compared with the original strain, their SW contents were significantly increased (*p* < 0.01) by 44.82%, 20.11%, 76.56% and 114.01%, respectively. The mutant with the highest content of SW in mycelia was 70. The average content of SW in mutated strains numbered 2 and 61 were 195.57 μg/g and 156.99 μg/g, respectively, which were significantly decreased (*p* < 0.05) compared with the original strain, and decreased by 10.88% and 28.46%, respectively. Mutant 61 was the strain with the lowest content of SW. 

#### 3.2.3. Observation of Mutant Colony Morphology and Stability Test of SW Production

As shown in Figure 5, all mutants in the liquid culture medium had obvious differences in culture medium color, shape and size of mycelium pellets compared with the original strain. To be specific, the liquid medium color of the original strain was orange-red, and the mycelium pellets were round and flesh-colored, with diameters of about 1 cm, and no mycelium bulge on the surface (Figure 5g,h). The liquid medium color of mutant 55 was yellowish brown, and the mycelium pellets were small and round and flesh-colored, with diameters of about 0.7 cm (Figure 5a). The liquid medium color of mutant 63 was light orange, and the mycelium pellets were grayish white in medium size, about 1 cm in diameter, with gray mycelium bulges on the surface (Figure 5b). The liquid medium color of mutant 64 was dark gray, and the mycelium pellets were black and similar in size to sago with diameters of about 0.5 cm (Figure 5c). The liquid medium color of mutant 70 was grayish brown, and the mycelium pellets were light brown with diameters of about 0.5 cm (Figure 5d). The liquid medium color of mutant 2 was orange, and mycelium pellets were white, with diameters of about 0.3 cm (Figure 5e). The liquid medium color of mutant 61 was light orange, and the shape of yellow-white mycelium pellets was irregular, with diameters of about 1.5 cm (Figure 5f).

As can be seen from Figure 6, the colony morphology of the original strain on PDA was larger, the colony base color was black, and the colony was gray to black (Figure 6h). The colonies of mutants 55 and 64 on PDA were white and the base color of the colonies was gray, among which the colony morphology of mutants 64 was larger (Figure 6a,b). The colony morphology of mutant 63 was medium in size, and its mycelium color and colony background color were similar to those of the original strain (Figure 6c). Both mutant 70 and 2 had smaller colony morphology, among which mutant 2 had the smallest colony morphology, and the colony color was similar to the original strain, but the growth was slower than the original strain (Figure 6d,e). After heavy-ion irradiation mutagenesis, the mutant strain 61 in the initial screening was brownish yellow in general, and the colony morphology was irregular convex in the middle with white mycelia in the overlying part. After subculture, the colony morphology of subcultured mutant 61 was larger, the mycelia were grayish white, and the base color of the colony was gray (Figure 6f,g).

**Figure 5 jof-08-00951-f005:**
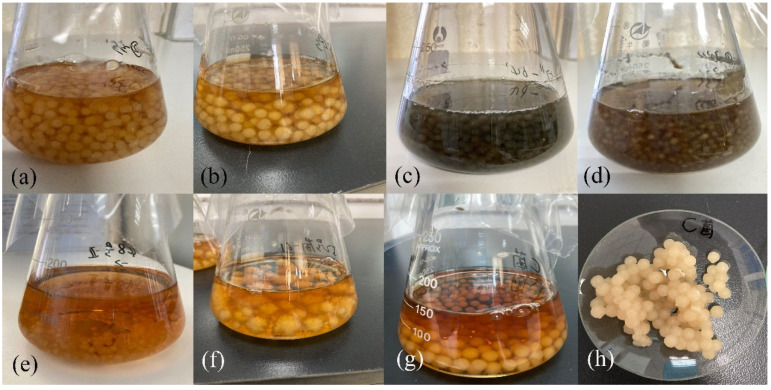
The mycelial morphology of strain in liquid medium. ((**a**) was mutant 55, (**b**) was mutant 63, (**c**) was mutant 64, (**d**) was mutant 70, (**e**) was mutant 2, (**f**) was mutant 61, (**g**,**h**) were original strains).

The mutants numbered 2, 55, 61, 63, 64 and 70 were continuously subcultured for 5 generations. After the cultivation, UPLC-MS/MS technology was used to detect the content of SW in mycelia, and four strains with stable SW production were screened, that were 55, 61, 64 and 70, respectively. The results are shown in Table 7.

### 3.3. Metabolomics Studies

#### 3.3.1. Data Quality Control

The correlation analysis results of QC samples are shown in Figure 7. The pairwise correlation R^2^ of QC samples were very close to 1, indicating the high correlation of QC samples, good stability of the whole testing process and high data quality.

PCA analysis results of total samples are shown in Figure 8. QC samples were distributed in a concentrated way with small differences, indicating good stability of the instrument and reliable test data. The sample points of A_61, A_C and A_70 had obvious separation in space, indicating that there were significant differences in metabolic patterns among the low-yielding SW mutant 61, the original strain and the high-yielding SW mutant 70.

#### 3.3.2. Differential Metabolite Screening

According to the threshold VIP > 1.0, FC > 1.2 or FC < 0.833 and *p* < 0.05, the number of differential metabolites screened is shown in Table 8, and a total of 843 metabolites were identified. Compared with the original strain A_C, 186 differential metabolites were screened out in low-yielding SW group A_61, including 171 metabolites significantly up-regulated and 15 metabolites significantly down-regulated. Compared with A_C, 251 differential metabolites were screened in high-yielding SW group A_70, among which 130 metabolites were significantly up-regulated and 121 metabolites were significantly down-regulated. 331 differential metabolites were screened out when A_61 compared with A_70, among which 254 metabolites were significantly up-regulated and 77 metabolites were significantly down-regulated. The Venn diagram intuitively compares the common and unique differential metabolites between different groups (Figure 9). The volcanic diagram (Figure 10) directly reflects the overall distribution of different metabolites in each group. Detailed information on differential metabolites in different comparison pairs is provided in Appendix A.

Clustering heatmap was used to visually analyze the clustering degree of different compound abundance levels between low-yielding SW group A_61, original strain group A_C and high-yielding SW group A_70. As shown in Figure 11, these three strains with different SW producing performance were clearly divided into three clusters, indicating that metabolomics data could distinguish the strain with low SW yield, the original strain and the strain with high SW yield. The significant differences in the abundance of these metabolites suggest that these metabolites could be used for enrichment analysis of the SW biosynthetic pathway in *A. oxytropis*.

#### 3.3.3. KEGG Enrichment Analysis

In order to study the biosynthetic pathway of SW in *A. oxytropis*, KEGG enrichment analysis was performed on the selected differential metabolites in each group. In comparison of A_61 and A_C, 27 pathways were enriched. A total of 47 pathways were enriched through the comparison of A_70 and A_C, and 14 pathways were significantly enriched (*p* < 0.05), including lysine degradation, glyoxalate and dicarboxylate metabolism, lysine biosynthesis, etc. Moreover, 47 pathways were enriched by comparing A_61 with A_70, among which ABC transporters, arginine biosynthesis and tryptophan metabolism were significantly enriched (*p* < 0.05). The top 20 pathways in each comparison pair are shown in Figure 12, Figure 13 and Figure 14.

## 4. Discussion

In the extraction process of natural secondary metabolites, the change of extraction time, extraction temperature, material-to-liquid ratio, solvent type and other factors will lead to changes in the yield of the substance to be extracted. Ultrasound-assisted extraction method was adopted in this study, which was simple in operation and low in cost. Through single-factor experiment, four factors, including time, temperature, material-to-liquid ratio and formic acid concentration, that had great influence on the extraction of SW in *A. oxytropis* were selected for optimization. The results of single factor and interaction of each factor indicated that the influence of these four factors on the extraction rate of SW was not in the same trend, and the increase after reaching a certain limit would lead to the decrease in the SW extraction rate. The results of the response surface methodology showed that the predicted value of SW extraction was 211.504 μg/g with the extraction time of 89.66 min, temperature of 41.36 °C, concentration of formic acid of 0.81% and material–liquid ratio of 1:24.96 g/mL. Guided by the parameters of response surface optimization, the optimal extraction conditions of SW in *A. oxytropis* were determined according to the actual operation conditions as follows: extraction time of 90 min, ultrasonic temperature of 41 °C, concentration of formic acid of 0.8%, and material–liquid ratio of 1:25 g/mL. The content of SW in *A. oxytropis* was 220.572 μg/g according to these extraction conditions, which was near the predicted value. In the determination of the SW extraction experiment in *A. oxytropis*, the extraction method, ultrasonic power, extraction times and the selection of extraction solvent all affected the determination of the SW content. The single-factor test in this study still lacked comprehensive consideration and few factors were selected. The factors affecting the extraction rate of SW could be further studied in the future.

In this study, *A. oxytropis* mutants with high and low SW yields were bred by heavy-ion irradiation. The results showed that heavy-ion irradiation led to the obvious mutagenesis of *A. oxytropis*, and the radiation lethality of *A. oxytropis* represented an upward trend. When the irradiation dose was 40 Gy, the average fatality rate was 69.23%; when the irradiation dose increased to 60 Gy, the average fatality rate decreased slightly to 66.67%, and then the fatality rate continued to rise. This was a phenomenon of low-dose hyper-radiosensitivity (HRS) and increased radioresistance (IRR). That is, when cells were damaged by external irradiation, they would start their own repair program, but the starting condition was that the irradiation dose must reached a certain threshold [23]. Therefore, it was speculated that when the cells received the irradiation dose of 60 Gy, they started their own repair pathway, resulting in a small decrease in the fatality rate. After the dose continued to increase, the cell damage was difficult to be repaired, resulting in cell death. Hao et al. obtained U4 and UD1 mutant strains with high SW content by ultraviolet irradiation and nitrosoguanidine treatment of *A. oxytropis*, and their yields increased by 16.02% and 21.87% compared with the original strain, respectively [24]. Four mutants were screened according to the content of SW in mycelia and the stability test of SW production after heavy ion beam irradiation. Among them, three strains had enhanced SW production performance, numbered 55, 64 and 70, and their SW production content increased by 44.82%, 76.56% and 114.01% on average, respectively. While one strain had weakened SW production performance, numbered 61, its SW production content was reduced by 28.46% on average. In addition, morphological results showed that the four mutants had significantly different phenotypic characteristics compared with the original strain either in liquid medium or on PDA solid medium. These results confirmed that heavy-ion irradiation had good mutagenic effect.

The mutant strains with different SW production performance obtained by heavy-ion irradiation could be used as the basis of metabonomics research to explore the biosynthetic pathway of SW in *A. oxytropis*, that had great significance. According to the results of KEGG enriched differential metabolic pathways, it could be found that most of these pathways were related to amino acid metabolism, indicating that SW biosynthesis was closely related to amino acid metabolism. Moreover, ABC transporters, galactose metabolism and tryptophan metabolism were all enriched and ranked by the front. ATP-binding cassette (ABC) is one of the largest known protein families, and its main function is to actively transport substrates by using ATP energy supply [25,26]. Elucidating the transport function of ABC transporters on SW or its biosynthesis related substrates would help to understand the molecular mechanism of SW biosynthesis and secretion pathway and provide important elements for the development of SW synthetic biology. According to KEGG enrichment results, ABC transporters were preliminarily predicted to be related to SW synthesis and accumulation, and might be responsible for SW secretion and related precursor transport. Transporter proteins played a key role in the process of transporting secondary metabolites produced by organisms to specific organelles to form subcellular compartments or secreted out of the body. Some alkaloids synthesized by certain organisms might also be potentially toxic to the cells themselves, so these alkaloids would be stored or secreted in specific organelles after synthesis and exerted their functions through transport proteins [27]. Transporters were closely related to the biosynthesis of secondary metabolites in organisms, and their expression levels generally showed the same trend as the substrate related synthases. At the same time, transporters could change the yield of metabolites synthesized by engineering strains, and their encoding genes were often located in the biosynthetic gene cluster of the target products. When the synthesis of the target product increased and could not be secreted normally or transferred to specific organelles for storage, the negative feedback mechanism would act at the synthesis site, thus changing its yield [28]. LUO et al. speculated there was a related secretion mechanism in the SW synthesizing fungus when studying *Metarhizium robertsii*, which might be related to *swnT* gene located in the SW synthesizing gene cluster “SWN” [29]. Therefore, it was speculated that the ABC transporters might alter the production performance of SW in *A. oxytropis* through the transport of SW biosynthesis related precursors and SW secretion. Galatose metabolism mainly produces glucose to participate in energy supply. Tryptophan metabolic pathway involves the metabolism of indoles, indoles acetate and other substances, but it is currently unknown whether there is a correlation between it and the synthesis of indoleisidine alkaloid SW, which requires further research.

Lysine degradation and lysine biosynthesis were closely related to SW biosynthesis in the KEGG pathway enriched by the above three comparison pairs. In high-yielding SW mutant strain 70, the low level of differential metabolites involved in lysine degradation pathway were saccharopine, L-pipecolate and cadavenine, while the high level of differential metabolites was L-2-aminoadipate, 2-oxoadipate and 5-phosphooxy L-lysine; the low level of differential metabolites involved in lysine synthesis pathway were 2-oxoglutarate, L-lysine and L-saccharopine, and the high level of differential metabolites were 2-oxodipate and L-2-aminoadipate. In low-yielding SW mutant 61 and the original *A. oxytropis* strain, the levels of these differential metabolites were reversed from mutant 70. The SW biosynthetic pathway in *A. oxytropis*, as shown in Figure 1, was speculated by combining the lysine degradation pathway in KEGG and the related biosynthetic pathways reported by previous researchers [30,31]. The high-yielding SW mutant 70 had higher levels of the precursors of lysine synthesis, such as L-2-aminoadipate and 2-oxodipate, while lower levels of L-lysine, saccharopine and L-pipecolate, leading to high SW yield. L-pipecolate generated 1-keto-indolicidine under the action of polyketide synthase (PKS), followed by the formation of SW, indicating that the synthesized lysine, saccharopine and L-pipecolate were consumed more. In contrast, the low-yielding SW mutant 61 had lower level of lysine precursor, but higher contents of L-lysine, saccharopine and L-pipecolate, as well as lower SW yield, indicating that these substances were less utilized, so the detection level was higher. Therefore, it was speculated that the different SW-producing properties of the mutants might be related to the biosynthesis and utilization of L-lysine and L-2-aminoadipate. It has been reported that SW biosynthetic pathway of *A. oxytropis* might have two branches, P6C and P2C, but the specific details and related catalytic enzymes remain unclear [32]. The proposed pathway of SW synthesis was consistent with the previous reports, which provided new details and clues for clarifying the SW biosynthesis pathway in *A. oxytropis*. 

## 5. Conclusions

The optimal conditions for ultrasonic-assisted extraction of SW from *A. oxytropis* optimized by response surface model were an extraction time of 90 min, ultrasonic temperature of 41 °C, formic acid concentration of 0.8% and the material–liquid ratio of 1:25 g/mL. The average yield of SW under this condition was verified to be 220.572 μg/g. Four mutant strains with significant and stable SW producing performance were screened after mutagenesis of *A. oxytropis* by heavy-ion irradiation, numbered 55, 64, 70 and 61, respectively. Their average contents of SW were 317.79 μg/g, 387.44 μg/g, 469.64 μg/g and 156.99 μg/g, that were 44.82%, 76.56%, 114.01% higher than that of the original strain, and 28.46% lower than that of the original strain. Metabolomics studies have found the related pathways affecting SW biosynthesis include lysine degradation, lysine biosynthesis, ABC transporters, etc. It was speculated the different SW production performance of the mutants might be related to the biosynthesis and utilization of L-lysine and L-2-aminoadipate.

## Data Availability

The data presented in this study are available within the article and the Appendix A files.

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
