# Peer review of "Screening of Endophytic Fungi in Locoweed Induced by Heavy-Ion Irradiation and Study on Swainsonine Biosynthesis Pathway"

_jof, 2022, doi:10.3390/jof8090951_

Round 1

Reviewer 1 Report

The manuscript presents an original study concerning the production of a particular fungal metabolite associated with different plants used in livestock feeding. The production of the metabolite is important as it manifests toxicity to animals and the availability of the compound for research is limited due to difficulty in the production and purification.

The authors proposed a strategy for selecting mutant strains (of Alternaria oxytropis) and have assessed the production of the metabolite with the obtained strains.

The design protocol is adequate for the purpose and all the results are presented and discussed in an appropriate manner.

The article can be published in the present form.

Author Response

Response to Reviewer 1 Comments

Dear Reviewer,

Thank you very much for your careful and comprehensive review of our paper. Your comments are a great encouragement to us! Thank you for your affirmation of our work. Thank you!

Comments and Suggestions. The manuscript presents an original study concerning the production of a particular fungal metabolite associated with different plants used in livestock feeding. The production of the metabolite is important as it manifests toxicity to animals and the availability of the compound for research is limited due to difficulty in the production and purification.

The authors proposed a strategy for selecting mutant strains (of Alternaria oxytropis) and have assessed the production of the metabolite with the obtained strains.

The design protocol is adequate for the purpose and all the results are presented and discussed in an appropriate manner.

The article can be published in the present form.

Response: Thank you for your comments and suggestions. We have read your comments carefully. Thank you again for your careful and rigorous review of the article and your affirmation of our experimental design. We will keep working hard in the future work. Thank you!

Reviewer 2 Report

The research topics and methodology used are very interesting, but the manuscript is prepared careless with numerous errors. This paper requires major revisions before it is ready for publication. Further detailed comments for consideration are provided below.

Comment 1#

Authors should use the new JoF Microsoft Word template file (2022).

Comment 2#

The list of authors ends with ... “and” ???? Correct it.

Comment 3#

Line 2: "sect. or section" instead of "Section" (improve anywhere)

Comment 4#

Keywords line 1 correct the space.

Comment 5#

In the text, reference numbers should be placed in square brackets [ ], and placed before the punctuation; for example [1], [1–3] or [1,3]. For embedded citations in the text with pagination, use both parentheses and brackets to indicate the reference number and page numbers; for example, [5] (p. 10). or [6] (pp. 101–105).

Comment 6#

Did the authors perform genetic identification of the tested strain? Do they have accession number?

Comment 7#

Line 72: Correct the name, Italic or not

Comment 8#

Line 73: remove italics

Comment 9#

Line 77: italics

Comment 10#

Line 83: correct unit abbreviation (rpm)

Comment 11#

Correct the units on the figure 1

Comment 12#

Poor quality of charts (figure 2)

Comment 13#

Correct the locations of Table 7

Comment 14#

Are the metabolic pathways an in-house study or based on available literature? I suggest either providing a citation or moving the diagram to results.

Comment 15#

Correct references according to journal guidelines.

Author Response

Response to Reviewer 2 Comments

Dear Reviewer,

I am very grateful to your comments for the manuscript. According with your comments and suggestions, we amended the relevant part in manuscript. All of the comments and suggestions are responded as follows. All the lines and pages mentioned in responses are form the revised manuscript in review mode (the new JoF Microsoft Word version).

Comments and Suggestions. The research topics and methodology used are very interesting, but the manuscript is prepared careless with numerous errors. This paper requires major revisions before it is ready for publication. Further detailed comments for consideration are provided below.

Response: Thank you for your comments and suggestions. We are sorry for the errors and we have revised them in the manuscript. Thank you!

Comment 1. Authors should use the new JoF Microsoft Word template file (2022).

Response: Thank you for your suggestion. We are sorry for this inattention. We have revised the original manuscript according to the new JoF Microsoft Word template file (2022). Thank you!

Comment 2. The list of authors ends with ... “and” ???? Correct it.

Response: Thank you for your suggestion. We are sorry for our negligence. We have revised it in the manuscript (page 1). Thank you!

Comment 3. Line 2: "sect. or section" instead of "Section" (improve anywhere)

Response: Thank you for your suggestion. We have revised it in the manuscript (Abstract, line 2; Keywords, line 1; Introduction, line 17; Materials and Methods, line 65; Abbreviations, line 594). Thank you!

Comment 4. Keywords line 1 correct the space.

Response: Thank you for your suggestion. We are sorry for the carelessness. We have revised it in the manuscript (Keywords, line 1). Thank you!

Comment 5. In the text, reference numbers should be placed in square brackets [ ], and placed before the punctuation; for example [1], [1–3] or [1,3]. For embedded citations in the text with pagination, use both parentheses and brackets to indicate the reference number and page numbers; for example, [5] (p. 10). or [6] (pp. 101–105).

Response: Thank you for your suggestion. We have revised these format mistakes in the manuscript. Thank you!

Comment 6. Did the authors perform genetic identification of the tested strain? Do they have accession number?

Response: Thank you for your comment and suggestion. The genetic identification of the tested strain hasn’t been performed and they do not have accession number yet. This is the work what we are going to do next. Thank you!

Comment 7. Line 72: Correct the name, Italic or not

Response: Thank you for your suggestion. We are sorry for our negligence. The name should be Solarbio Science & Technology Co. Ltd. (Beijing, China). We have revised it in the manuscript (page 3, lines 72-73). Thank you!

Comment 8. Line 73: remove italics

Response: Thank you for your suggestion. We are sorry for our negligence. We have revised it in the manuscript (page 3, line 73). Thank you!

Comment 9. Line 77: italics

Response: Thank you for your suggestion. We are sorry for the carelessness. We have revised it in the manuscript (page 3, line 77). Thank you!

Comment 10. Line 83: correct unit abbreviation (rpm)

Response: Thank you for your suggestion. We are sorry for this inattention. We have revised it in the manuscript (page 3, line 83). Thank you!

Comment 11. Correct the units on the figure 1

Response: Thank you for your comment and suggestion. We have revised the units of Figure 1 (page 7, lines 251-252). Thank you!

Comment 12. Poor quality of charts (figure 2)

Response: Thank you for your comment and suggestion. We are sorry for the poor quality of Figure 2 (page 9, lines 277-278). We have revised it. Thank you!

Comment 13. Correct the locations of Table 7

Response: Thank you for your suggestion. We have revised it in the manuscript (page 15, line 394). Thank you!

Comment 14. Are the metabolic pathways an in-house study or based on available literature? I suggest either providing a citation or moving the diagram to results.

Response: Thank you for your comment and suggestion. We are sorry for the unclear description. The SW biosynthetic pathway in A. oxytropis. was speculated by combining available literatures [30, 31] and the lysine degradation pathway in KEGG (https://www.kegg.jp/pathway/map00310). We have revised relevant part in the manuscript (page 22, lines 552-555) to make the description clearer. Thank you!

  1. Lu, H.; Quan, H.; Ren, Z.; Wang, S.; Xue, R.; Zhao, B. The genome of undifilum oxytropis provides insights into swainsonine biosynthesis and locoism. Sci Rep. 2016, 6, 30760.
  2. Tan, X.M.; Chen, A.J.; Wu, B.; Zhang, G.S.; Ding, G. Advance of swainsonine biosynthesis. Chinese Chem Lett. 2018, 29, 417-422

Comment 15. Correct references according to journal guidelines.

Response: Thank you for your suggestion. We are sorry for our negligence. We have revised references according to journal guidelines. Thank you!

Round 2

Reviewer 2 Report

The suggested corrections have been made in the manuscript, however, the JoF Microsoft Word template file is still old (2021). Please correct this.